# Cancer Vaccines in the Immunotherapy Era: Promise and Potential

**DOI:** 10.3390/vaccines11121783

**Published:** 2023-11-29

**Authors:** Chaitenya Verma, Vishakha Anand Pawar, Shivani Srivastava, Anuradha Tyagi, Gaurav Kaushik, Surendra Kumar Shukla, Vinay Kumar

**Affiliations:** 1Department of Pathology, Wexner Medical Center, Ohio State University, Columbus, OH 43210, USA; chaitenya999@gmail.com; 2The University of Texas MD Anderson Cancer Center, Houston, TX 77030, USA; vishakha.pawar284@gmail.com; 3Department of Pathology, School of Medicine, Yale University, New Haven, CT 06510, USA; shivani.srivastava@yale.edu; 4Department of cBRN, Institute of Nuclear Medicine and Allied Science, Delhi 110054, India; anuradhatyagi07@gmail.com; 5School of Allied Health Sciences, Sharda University, Greater Noida 201310, India; gaurav.kaushik@sharda.ac.in; 6Department of Oncology Science, OU Health Stephenson Cancer Center, Oklahoma City, OK 73104, USA; 7Department of Physiology and Cell Biology, The Ohio State University Wexner Medical Center, Columbus, OH 43201, USA

**Keywords:** cancer vaccine, immunotherapy, tumor microenvironment, immune checkpoint, tumor cell, immune suppression

## Abstract

Therapeutic vaccines are a promising alternative for active immunotherapy for different types of cancers. Therapeutic cancer vaccines aim to prevent immune system responses that are not targeted at the tumors only, but also boost the anti-tumor immunity and promote regression or eradication of the malignancy without, or with minimal, adverse events. Clinical trial data have pushed the development of cancer vaccines forward, and the US Food and Drug Administration authorized the first therapeutic cancer vaccine. In the present review, we discuss the various types of cancer vaccines and different approaches for the development of therapeutic cancer vaccines, along with the current state of knowledge and future prospects. We also discuss how tumor-induced immune suppression limits the effectiveness of therapeutic vaccinations, and strategies to overcome this barrier to design efficacious, long-lasting anti-tumor immune responses in the generation of vaccines.

## 1. Introduction

The smallpox vaccine introduced by Edward Jenner’s ground-breaking article in 1798 opened new avenues in the field of advanced immunology [1]. Later, Dr. Willian Coley first administered Coley’s Toxin, an inactivated mixture of *Streptococcus pyogenes* and *Serratia marcescens*, to a cancer patient in 1891 in an effort to boost the immune system and improve the patient’s health. The immune system attacked the patient’s tumor, causing it to disappear. In the next 40 years, he continued to treat hundreds of soft-tissue sarcoma patients using this immunotherapy and became the “Father of Immunology”, inspiring millions of scientists and clinical scientists [2]. Based on research on vaccines over two centuries, twenty-six infectious diseases are preventable through vaccination today. However, despite many efforts, a number of other bacterial, viral, and parasitic illnesses continue to defy vaccination protection. The chronological progression of several cancer vaccines’ approval is illustrated in Figure 1.

Despite extensive research and development, it has been difficult to design and utilize cancer vaccines as effective therapeutics in the clinic. The U.S. Food and Drug Administration (FDA) has nevertheless approved two preventative vaccines—one for the hepatitis B virus, which is responsible for liver cancer, and another for the human papillomavirus, which causes cervical cancer. Immunotherapy for cancer encompasses an extensive spectrum of modern and innovative potentially life-saving treatments aimed at eliminating tumors by activating host anti-tumor immunity [3]. FDA approval of immune checkpoint blocking (ICB) therapies and chimeric antigen receptor (CAR)-engineered T cell immunotherapies has yielded substantial advancements in the field of cancer treatment. However, it is important to note that these therapies may not exhibit optimal efficacy in all individuals diagnosed with cancer, and their response varies among different types of cancer [4]. Cancer vaccines, although they have not yet demonstrated a comparable clinical impact, possess significant potential for both preventive and therapeutic purposes. They can be utilized as single agent or in combination with other therapeutic approaches [5,6], and have the ability to confer long-lasting immunity against cancer recurrence. Consequently, they hold promise as a crucial component of future combinatorial immunotherapies [7,8]. Therapeutic cancer vaccines are given to cancer patients with the goal of eliminating cancer cells by augmenting the patient’s own immune responses, as opposed to preventive vaccines, which are given to healthy individuals [9,10].

The term “vaccination” is now expanded beyond that, to refer to interventions that target a disease’s specific antigen in order to treat or alleviate the existing pathology of the disease. In addition to triggering a new immune response in immunocompromised or naïve individuals, vaccinations can boost existing immunity and alter its type to better combat the targeted diseases via systemic versus mucosal and Th1 versus Th2 responses [8,11]. The objective of this review is to provide a thorough examination of recent developments in research in the realm of therapeutic cancer vaccines. Additionally, it seeks to assess the present state of this field, identify the challenges that it faces, and outline possibilities for the development of novel cancer vaccines as a promising therapeutic strategy in the future.

## 2. Cancer Vaccines

### 2.1. Therapeutic Cancer Vaccine

Cancer therapeutic vaccines can be classified as either cell-based or antigen-based. Cell transfer is used in cell-based vaccinations to elicit targeted anti-tumor immunity [12,13]. Some examples of whole-cell vaccines are adoptive T-cell transfer, in which healthy T cells are transferred to a patient, and allogeneic (same-species) cancer cells. On the other hand, vaccines based on antigens intend to improve the ability of the body to defend itself by introducing a new antigen or a different way of presenting an existing one [14,15]. Types of cancer vaccines are summarized in Figure 2.

DNA and mRNA cancer vaccines are an effective and adaptable form of immunotherapy. Humoral and cellular immunity can be induced by these cancer vaccines due to the encoding and expression of tumor-associated antigens (TAAs), tumor-specific antigens (TSAs), and related cytokines. Engineered DNA molecules, either alone or in combination with other immunomodulatory substances, that encode one or more designated TAs [16], are the basis of cancer DNA vaccines. In order to activate transcription, a DNA vaccine must enter the cytoplasm of antigen presenting cells (APCs), cross the cell membrane, and make its way to the nucleus. The resulting mRNAs go into the cytoplasm to be translated into TAs. These proteins can undergo degradation by proteasomes and undergo intracellular processing within the endoplasmic reticulum, resulting in the generation of intracellular antigens. These antigens are then presented as peptide-MHC-I complexes. However, the genetic information for TAs is also delivered via mRNA vaccines. They are created by in vitro transcription of template DNA using RNA polymerase [17,18]. Tumor-derived mRNA can be amplified by polymerase chain reaction (PCR) to produce a significant quantity of complementary DNA encoding patient-specific TAs.

The delivery of DNA and mRNA vaccines into the cell is the most challenging aspect of these therapies. These vaccinations have to get through the cell membrane, which is a lipid bilayer containing zwitterionic negatively charged phospholipids, ion pumps, and channels that all work to keep the cell at a negative potential. In addition, DNA and RNA sensors and endonucleases form an intercellular barrier. The vaccine is taken up and processed by endocytic pathways before being released into the cells once it reaches the plasma membrane. Protonation of the lipid nanoparticle (LNP) residual amines disrupts the endosomal membrane, allowing a small percentage of LNPs to escape the endocytic route [19,20]. This causes the LNP cargo to be released into the cell too soon, reducing the vaccine’s efficacy. The success of DNA and mRNA vaccines relies heavily on the mechanism of administration. Vaccine efficacy and safety are influenced by the anatomical and physiological characteristics of the immunization location [17], which can be the skin, a lymphoid organ, or muscle. Systemic delivery is one of the strategies when the vaccine is injected into the bloodstream (intravenously) to reach and influence immune cells in the body [21]. However, vaccines can also be delivered directly to the site of action by local injections (subcutaneously). The potential for systemic delivery-related adverse effects is diminished when the drug is administered subcutaneously. DNA and mRNA vaccines are often administered via intramuscular injection. The vast vascular network that makes up muscles brings a wide variety of immune cells, including APCs, to the injection site [22]. DNA and mRNA vaccines can also be injected intradermally or intranodally, where APCs and other immune cells can quickly interact with and ingest them [23,24].

Moreover, proteins or DNA can be injected directly to stimulate the immune system, or they can be delivered via a vector to prevent degradation and further stimulate the immune system [25,26]. These vectors include attenuated bacteria (Salmonella, Mycobacterium, Listeria, or Shigella), yeast (nonpathogenic Saccharomyces cerevisiae), and modified viruses (adenovirus or poxvirus, with the vaccinia virus being the most common). All the bacteria mentioned here are intracellular and can directly target antigen-presenting cells (APCs), interfering with different antigen-presenting pathways (MHC class I or MHC class II) depending on where in the APC they are located [27,28].

Targets aimed at tumor inoculations can be categorized into two groups: one is tumor-associated antigens (TAAs) and the other is tumor-specific antigens (TSAs). TAAs as, or in the form of, self-antigens could be selectively or peculiarly expressed in cancer cells. On the other hand, TSAs, encompassing neoantigens as well as onco-viral antigens, are tumor specific and have been recognized in onco-viral cancers such as human papillomavirus (HPV)-linked cervical cancer and human herpesvirus 8-linked Kaposi sarcoma [29]. These neoantigens (NeoAgs) are formed as a byproduct of somatic mutations developed through carcinogenesis. The NeoAgs that are shared among patients are commonly observed across different individuals, and their presence is determined by oncogenic driver mutations. On the contrary, the mainstream of NeoAgs is exclusive to singular patients’ tumors (private NeoAgs) [30]. With the advancements in sequencing technologies, the extrapolation of major histocompatibility complex (MHC)-tied epitopes and their enrichment strategies enable steering of tumor NeoAg assortment on the specific patient of interest [31].

Cancer stem cells (CSCs) have been contemplated as potential therapeutic targets for cancer treatment because of their possession of self-renewal capability, etc. The regulation of various biological activities of CSCs is directed with the help of widely known pluripotent TFs, such as Sox2, Nanog, KLF4, OCT4, and MYC. Along with these regulators, various signaling intracellular networks of JAK-STAT, NF-κB, Hedgehog, Notch, Wnt, PI3K/AKT/mTOR, TGF/SMAD, and PPAR influence CSCs. Other crucial regulators are extracellular factors, in the form of cancer-associated mesenchymal stem cells, tumor-associated macrophages, cancer-associated fibroblasts, hypoxia, exosomes, and extracellular matrix [32]. Recent reports have confirmed the efficacy of the CSC-DC vaccine in inhibiting the metastasis of primary tumors and stimulating the humoral immune reactions against CSCs [33]. Programmed death ligand 1 (PD-L1) enhances the persistence of tumor-reactive T lymphocytes against cancer cells by binding to programmed death-1 [34,35]. Furthermore, both cell-based and antigen-based vaccines, as described in Table 1, are being trialed or under development for their respective target cancer types.

### 2.2. Cell-Based Vaccines

In 2010, the first therapeutic cancer vaccine, Sipuleucel-T (ProvengeTM), was licensed by FDA for clinical use in prostate cancer treatment [44]. Patients with symptomatic illness were shown to have a 4.1-month median survival benefit and an 8.7% prolonged 3-year survival when treated with Sipuleucel-T compared to placebo [45,46]. Metastatic castration-resistant prostate cancer is treated with the vaccination, which is an autologous active cellular immunotherapy. APCs, such as dendritic cells (DCs) and macrophages, are isolated from the patient’s peripheral blood and then activated in vitro with a recombinant fusion protein (PA2024) of prostatic acid phosphatase and granulocyte macrophage colony stimulating factor (PAP-GM-CSF). Immunostimulatory granulocyte macrophage colony-stimulating factor (GM-CSF) is fused to a prostate antigen (prostatic acid phosphatase) in this fusion protein. The reintroduced activated APCs will subsequently stimulate a CTL immunological response against the prostate tumor cells. Sipuleucel-T is administered as a 60 min infusion every two weeks for three doses, and each dosage consists of a minimum of 50 million activated autologous CD54+ cells [38,47]. Adoptive T-cell transfer is another promising method for inducing an anticancer CTL response. This allows in-vitro selection, generation, and activation of a large number of anticancer T-cells that detect tumor antigens prior to their reintroduction into the host [39,40]. Clinical trials of allogeneic whole cell vaccines have also shown promise. The goal of this treatment is to stimulate an immune response against the numerous antigens released by cancer cells by injecting irradiation of entire cancer cells from another host with the same tumor type into the patient. Since whole tumor cells are not very immunogenic, other immune-stimulating chemicals must be included in the vaccine formulation [36,37].

### 2.3. Autologous Tumor Cell Vaccine

Cancer vaccines that are made from the patient’s own tumor cells are called autologous tumor vaccines. Typically, these tumor cells are irradiated, mixed with an immunomodulatory adjuvant, and then given back to the patient from whom they were originally isolated [48,49,50]. Cancers including lung cancer, colon cancer, melanoma, kidney cancer, and prostate cancer have all been studied using autologous tumor cell vaccines [49,51,52,53,54]. It is possible to confer immunostimulatory properties on autologous tumor cells by genetic modification. These engineered tumor cells are safe and have a positive effect on patients’ by generating anti-tumor immune memory cells [55]. Via the mechanism of the autologous tumor cell vaccine, immunization with engineered tumor cells leads to the production of IL-2, a crucial cytokine that promotes Th1 immunity, as well as the development of a strong tumor suppression impact via high IFN-γ production and elevated NK cells and cyto-toxic cells. This is the mechanism by which autologous tumor cell vaccines work to prevent tumor growth. Immunization with engineered tumor cells also leads to the production of IL-2 [56]. For example, multiple tumor models, including ESb lymphoma and B16 melanoma, have demonstrated that autologous tumor cells infected with Newcastle disease virus produce tumor protective immunity [57]. Another extensively investigated vaccination, GM-CSF transduced autologous tumor cells vaccine (GVAX), recruits DCs for antigen presentation and priming cytotoxic T cells. Additionally, it activates macrophages, DCs, and NKT cells [58,59]. Overall, autologous cell vaccines were associated with improved and disease-free survival with greater expectancy [60].

### 2.4. Allogenic Tumor Cell Vaccine

Allogenic tumor vaccines are quite similar to autologous vaccines except that they use material from a different individual of the same species. Tumor-associated antigen (TAA) cell lines, which have been generated in the lab and are specific to a given tumor type, are a common source of allogenic material [15]. Vaccines made using these methods produce effective and safe immunogenic agents, with no risk of proliferation of the injected tumor cells [61]. The machinery behind the allogenic tumor cell vaccine is also very similar to that of autologous cell therapy. Engineered TAA cells produce IL-2 and promote Th1 immunity, and lead to IFN-γ production and NK and cytotoxic cell growth. To further facilitate the induction of an anti-tumor-specific immune response, irradiated cells naturally express and present a large number of TAAs, eliminating the need to identify and isolate TAAs [62,63]. Due to the fact that many allogenic cell-based vaccines are derived from cancer cell lines of the same species and type, they may not be able to detect tumor antigens that are unique to the individual patient [64]. Despite this drawback, many studies have shown increased immunogenicity in prostate cancer and aggressive melanomas [65,66].

### 2.5. Dendritic-Cell-Based Vaccine

DCs are specialized antigen-presenting cells that can activate both naive CD4 and CD8 T cells [67,68]; they play an important role in the initiation and regulation of innate and adaptive immune responses [69]. Multiple DC-targeted antigen clinical trials have demonstrated the safety and variable immune response of DC-targeted vaccinations administered in vivo [70,71]. Using poly-ICLC and/or resiquimond as an adjuvant, a trial found that a human anti-DEC-205 monoclonal antibody fused with the tumor antigen NY-ESO-1 produced a humoral and NY-ESO-1-specific CD4 and CD8 cells response, leading to partial clinical responses without toxicity [72]. Data from combinatorial therapy with DC vaccines are promising, such as blocking of the immune checkpoint (e.g., anti-CTLA4); additionally, the antigen delivery method also impacted vaccine outcomes. Improvements in nanoparticle or viral vector-based antigen targeting or delivery for DCs are a key factor in the expansion of DC-based vaccinations into clinical practice [73,74].

### 2.6. Antigen-Based/Protein/Peptide-Based Vaccine

Cancer vaccines in this category are developed by focusing on epitopes on peptides that can stimulate humoral and cellular immune responses against TAAs or TSAs. Peptide-based cancer vaccines have the capability of enhancing the effector adaptive immune response and provide persistent acquired immunity against a tumor antigen that is viewed as “foreign”. The ability to differentiate cancer cells from normal cells is facilitated by the upregulation or overexpression of endogenous proteins, as well as modifications in these proteins. Therefore, mutated or differentially expressed proteins in cancer cells could be used as therapeutic vaccine targets. Included in this group of antigens are cancer/germline antigens [75] and cell lineage differentiation antigens [76,77], both of which are uncommon in mature tissues. TAA expression in normal cells raises the probability that these cells will go through an immunological tolerance process, reducing their immunogenicity. However, TSAs, which are antigens caused by nonsynonymous mutations or other genetic alterations in cancer cells, stand out by not being expressed on the surface of normal cells. In addition, various studies have shown that TSAs vary widely depending on the kind of tumor [78,79]. APCs regularly present tumor antigens to adoptively transferred B and T lymphocytes. MHC-I molecules on APCs present peptides to CD8+ T cells, while MHC-II molecules on APCs present peptides to CD4+ T cells [80]. Activated CD8+ T lymphocytes are thus capable of recognizing cancer antigens expressed on the surface of tumor cells. Apoptotic molecules such as Perforin, Fas Ligand, and Granzymes are secreted in response to this identification, resulting in cell-mediated cytotoxicity [81]. Therapeutic vaccination against tumors relies on eliciting strong and long-lasting responses from CD4+ and CD8+ T cells, which can only be accomplished by delivering large quantities of highly immunogenic antigens to APCs. Additionally, it is vital to allow the infiltration of the tumor microenvironment (TME) by these T cells, while also assuring the durability and maintenance of the immune response. Several laboratories are currently investigating methods to optimize antigen presentation by activating and maturing APCs to stimulate T cells to respond in the best possible way. Focusing on new adjuvants that can stimulate and augment the strength and durability of the immune response mediated by antigen-specific T and B cells is the major goal of these approaches [82]. Different therapeutic peptide-based vaccination formulations have been tested in a range of tumor types during the past few decades. However, prior research has only found small effects, leading to a substantial clinical benefit [29,83].

Furthermore, vaccination against primary or recurrent noninvasive papillary carcinoma and/or invasive subepithelial connective tissue papillary tumors used a whole bacterial vaccine. TheraCysTM contains Bacillus Calmette-Guérin (BCG), which is a live attenuated strain of Mycobacterium bovis and serves as its active component [84]. Bacteria are introduced into the bladder intravenously, stimulating the immune system in response to the bacterial infection [85,86]. Mice were vaccinated with a recombinant DNA vaccine called pDERMATT (plasmid DNA Encoding Recombinant MART-1 and Tetanus toxin fragment-c) that contained an immunostimulatory tetanus toxin fragment-c and a DNA plasmid encoding the melanoma-associated antigen “melanoma antigen recognized by T-cells” (MART-1) [26]. Carbohydrate-based vaccinations are another method used; these vaccines involve a protein carrier fused to tumor-associated carbohydrate antigens. The goal of this approach is to use a foreign protein carrier to expose the self-antigen nature of the carbohydrate antigens associated with tumors to the immune system [87]. Relapsed prostate cancer patients were included in a phase I clinical trial to assess the safety of a tumor-associated carbohydrate Ag-KHL (antigen-Keyhole limpet hemocyanin) vaccination administered with a saponin immunologic adjuvant [42,88,89]. The study demonstrated that prostate-specific antigen (PSA) levels went down, as well as anti-tumor antibody titers [89]. Carbohydrates on tumor cells are the focus of current vaccination research [90]. The majority of therapeutic antigen-based cancer vaccines work by stimulating CD8+ lymphocytes (CTLs) to mount an anti-tumor cellular immune response [91]. It is common for vaccine-induced cellular immune responses to fall short because of insufficient amounts of high avidity immune cells in circulation. These cells need to not only reach the tumor, but also be activated properly before they can do any real damage to the tumor [92].

## 3. Clinical Landscape of Cancer Vaccines

Since 1890, when Coley found that streptococcal bacterial culture served as the major antagonist in some sarcoma instances, immunotherapy has undergone a dramatic transformation [93,94]. Based on the tumor regression mechanisms, the forms of cancer immunotherapy have been categorized. Simply put, these therapies could have either an active or a passive mode of action in a subject’s immunity [95]. In this regard, therapeutic cancer vaccines have endured a renaissance over the passing decades. An insight into tumor-corresponding antigens, an innate and adaptive immune response, and expansion of state-of-the-art technologies for antigen release have accelerated the development of better-quality vaccine design [96]. The ultimate objective of these vaccines is to encourage tumor regression, ascertain durable antineoplastic memory, eliminate minimal residual disease, and avoid unfavorable outcomes. Centered on these intents, numerous therapeutic cancer vaccines have been assessed in clinical trials. In the present review, a panorama of clinical studies with respect to cancer vaccines is presented to provide a better understanding of progressions in cancer treatment.

One of the first clinical studies for vaccine development was established from patient’s tumor cells. These vaccines were effective in various phases of cancer clinical studies due to an extensive antigen-specific reaction against tumor cells [55,96]. Bastin et al. presented a meta-analysis of clinical trials containing ~700 patients with a minimum one dose of vaccine with an overall safe tolerance level, with the exception of only five grade III events [60]. Haas et al. reported an escalation in immunogenicity of vaccines caused by a virus infection of non-virulent strain NDV Ulster and addition of bispecific antibodies [97].

One type of extracellular vesicles is the exosome. The research works of Pan et al. and Harding et al. provide the earliest evidence regarding exosomes’ crucial roles in cell signaling and their capabilities in revealing the characteristics of the cells that produce or discharge them during the crosstalk among various sites of the cells [98,99,100,101]. In 1986, Schirrmacher et al. characterized the role of tumor-derived exosomes (TDEs) in exhibiting the presence of antigens that were specific to their corresponding metastatic lymphoma and its variants [102]. These discoveries disclose the potential of exosomes as biomarkers for vaccine development [103,104]. In 2022, Huang et al. demonstrated the construction of an in situ DC vaccine (HELA-Exos) by packing the immunogenic cell death (ICD) inducers into breast-cancer-derived exosomes [105]. These in situ vaccines exhibited compelling anti-tumor effects in a mouse xenograft model of triple negative breast cancer (TNBC), as well as in patient-derived organoids by stimulation of cDC1s and CD8+ T cell activity against tumor cells [105]. Meng et al. reported utilization of murine ESC engineered exosomes to construct the granulocyte-macrophage colony stimulating factor vaccine for inhibiting murine lung cancer [106].

The central factors of antigen vaccine attainment are dependent on the nature of antigens, e.g., MHC-I or MHC-II restricted, the dosage of immunogen, the kind of adjuvant used, and the method of administration [96]. As shown in Table 2, DNA vaccines carry intrinsic adjuvants and are enriched with tumor-associated antigens. However, compared to their counterparts, DNA vaccines need additional stages of transcription and translation before the antigen cross-presentation for initiation of CD4+ and CD8+ responses [107,108]. A case study showed a DNA-based immunotherapy wherein pTOP plasmids encoded viral glycoprotein and harbored diverse extraneous T cell tumor epitopes, to generate immune detection by suitable dispensation of both MHC-I and MHC-II epitopes for anti-tumor activity in various tumor models [109]. A review article by Lopes et al. provides a comprehensive perspective regarding the current states of cancer DNA vaccines [110]. A phase II trial study is now being undertaken to elucidate the efficiency of the pTVG-HP DNA vaccine and its synergistic effects with pembrolizumab in curing castration-resistant prostate cancer [111].

Additionally, the time needed to produce each type of vaccine and the effectiveness of each type of approach is directly proportional to (a) the prompt scientific/technical progressions, (b) the documentation of well-defined populations that might profit from CV courses, (c) determinations to allow for comparison of diverse clinical examinations, and (d) the formation of a worldwide workforce that can maintain the potential requirements and supply chain [30]. Overall, there is growing enthusiasm for widespread vaccination operations, increased production developments, and, essentially, clinical outcomes from phase II/III experiments, which will illuminate the definitive role of CVs in cancer management in the subsequent years.

RNA vaccines are distinct from DNA ones with respect to their proximity towards protein antigen expression and APCs [96]. Their ability to be translated in the cytoplasm of the host system averts the possibility of oncogene activation because of its inability to amalgamate inside the host genome [112]. In 1996, the first mRNA cancer vaccine report described the potential APC properties of DC when pulsed with RNA, both in vitro and in vivo [113]. Subsequently, advancements in technologies have led to stable mRNA structural properties and better-quality delivery methods. Multiple clinical trials are now enrolling patients with cancer for mRNA-based vaccine treatments (Table 3). In 2017, Sahin et al. introduced the perception of mutanome vaccines, which can be directed against individual mutations by executing the RNA-involving poly-neo-epitope method [114]. In 2023, leading biotech companies such as Moderna and Merck declared FDA’s title of breakthrough therapy for MRNA-4157/V940, an RNA-based vaccine, synergized with KEYTRUDA(R), for the adjuvant care of subjects with high-risk melanoma. These companies are commencing a phase 3 clinical trial in 2023 with other tumor types (https://investors.modernatx.com). Although mRNA therapeutics has been advancing in vaccine development, more technological progression is required in the delivery of mRNAs that code for lethal intracellular proteins that carefully cause cell death in unsought and diseased cells [115,116]. In this light, several scientific reports demonstrate the efficiency of recruited cellular microRNAs (miRNAs) in targeting highly selective cell forms and regulating explicit disease expression profiles. This capability of miRNAs helps to inhibit protein expression from the mRNA in inadvertent recipient cells by manipulating endogenous pools of miRNA; apoptosis of tumor cells can thus be directed while preventing damage to the other cells [117]. However, a comprehensive functional classification of miRNAs of interest is essential for their successful therapeutic application. A thorough examination and confirmation of the authenticity of an miRNA is crucial to develop miRNA target-prediction algorithms that could enable accomplishment of experimental approaches to validate a higher quantity of targets [118,119,120,121].

Cancer vaccines categorized based on peptides contain a sequence of amino acids derived from TSAs or TAAs (Table 4) [122]. Their efficiency can be confirmed by the presence of CD8+ epitopes that would eventually trigger CTL anti-tumor immunity consequences, alongside CD4+ epitopes aimed at T-helper cell stimulation [123]. Hence, a robust immunogenic reaction depends on the sequence length of peptide vaccines. A longer peptide sequence grants a wider population exposure of HLA-types [124,125]. A ground-breaking work of Wu et al. confirmed the safe and beneficial usage of synthetic long peptide-based vaccines in clinical trials [126]. Their phase I clinical study reported no relapse of melanoma post-treatment of the 20 bp long slp-based vaccine, NeoVax, in four out of six melanoma subjects [127].

## 4. Cancer Vaccines in Combination Therapy

Therapeutic cancer vaccines are a special type of therapy because they start a dynamic process of triggering the host immune system, which can then be used by further therapies that are given concurrently or afterwards. Both in the clinical context and in preclinical models, the addition of immunotherapy to conventional cancer treatments has demonstrated effectiveness [128]. The immunosuppressive tumor microenvironment had a role in the cancer vaccines’ inability to mediate sustained regression of tumors in several of the clinical investigations. According to preclinical and early clinical studies, monotherapy is less effective than combining chemotherapy or checkpoint blockade with therapeutic cancer vaccines [129,130,131]. Therapeutic cancer vaccines promote immune cells’ infiltration into the tumor microenvironment (TME) and cytotoxic immune cell activation, whereas immune checkpoint inhibitors (ICIs) stop and/or reverse the immune cells’ dysfunction [132]. Chemoradiation therapy (CRT) has been used often to treat patients with unresectable esophageal squamous cell carcinoma (ESCC). Nevertheless, not all patients are chemoradiotherapy responsive, and many experience relapses. As of now, cancer vaccines have demonstrated excellent therapeutic outcomes and a tolerable safety profile for esophageal cancer (EC). As a result, treating EC with chemoradiotherapy and cancer vaccines may be successful [133] (Table 5).

## 5. Challenges and Promises in Cancer Vaccine Development

The poor immunogenicity of tumor antigens and tumor immune evasion mechanisms make the development of cancer vaccines challenging [140]. One of the major obstacles to the development of a successful cancer vaccine is the targeting of tumor antigens that may have low immunogenicity in the tumor environment or that could mutate to avoid the immune response [141]. The use of messenger RNA (mRNA) in cancer immunotherapy is growing in popularity because it can act as an effective vector for delivering therapeutic antibodies on immunological targets. Compared to conventional vaccines, mRNA vaccines have many benefits, including high efficacy, reduced toxicity, quick manufacture, and safe administration. The broad implementation of this method has been shadowed by the inefficient and instable delivery of mRNA. The first technique used to increase stability of mRNA is the modification of the 5′ cap. Second, immunological recognition of mRNA can be hindered by post-translational modifications of the mRNA [142].

The development of peptide-based vaccines faces several difficulties, though. The fact that the spatial configuration of the epitopes changes as the antigen attaches to cell surface receptors limits the accuracy and sensitivity of the T cell epitope prediction algorithms despite their widespread use [143,144]. False-positive and false-negative results may happen as a result [145]. The failure to elicit the necessary immunogenicity is a potential downside of DNA vaccines, and the primary causes here are insufficient DNA transfection and immunostimulation. Due to the complexity of the individual’s cellular and nuclear membranes, the DNA transfection capacity of these vaccinations produces non-uniform effects. Through endocytosis or pinocytosis, the plasmids must pass through the phospholipid-rich cell membrane. Furthermore, to avoid being degraded by nucleases, endosomes, and lysosomes, plasmids are necessary. By optimizing plasmid delivery through physical and chemical methods, these difficulties could be overcome [146]. Tumor targeting bacteria are a perfect vehicle for delivering therapeutic cargo that is selectively targeted at cancers of different origins because of their peculiar distinguishing traits, which include tumor selectivity, targeting the hypoxic environment of tumors, and unique gene packaging. However, despite the fact that modified bacteria have a great therapeutic potential to target tumors, a single anti-cancer agent might not be capable of treating a patient on its own due to the significant molecular and histologic heterogeneity of cancers. The small half-life of the bacterial protein peptide and the unsteady DNA present still another significant hurdle in this field [147]. One of the main restrictions of bacterial-based cancer therapy (BBCT) is that some kinds of chemotherapy might cause the immune system to become so suppressed that it is unable to adequately respond to colonization of bacteria. Additionally, live bacterial products can colonize in foreign bodies such as artificial heart valves, joint replacements, and implanted medical devices, which could act as infection reservoirs. Additionally, before cancer cells are penetrated, mutations in recombinant plasmids carried by bacteria can change the fate of anti-tumor action. There are a number of dangers that can result from this, such as therapy failure, infection, or death [148].

Combination adjuvant techniques are being developed and will be refined as a potential solution to overcome self-tolerance and tumor evasion mechanisms in order to trigger a strong anti-tumor response in opposition to these difficulties [141].

## 6. Conclusions

Most of the vaccines for the different types of cancer are either under clinical trial or still in the pre-clinical stage. In this review, we discussed the different types of vaccines and the current status of different vaccines that are under clinical trials for various types of malignancies. Different combination therapy approaches, along with potential cancer vaccines and the importance of chemotherapy and radiation therapy in vaccine responsiveness, were also discussed in detail. We concisely explored different state-of-the-art types of cancer vaccines, ranging from cell-based to mRNA-based cancer vaccines, in this review. Additionally, we also discussed the roadblocks and different types of challenges in the area of cancer vaccine development. Although significant progress has been made in the area of cancer vaccines, identification of novel antigens and simultaneous targeting of multiple tumor antigens may be the better strategy for vaccine development. With advancement of immunoinformatic approaches and better understanding of tumors as well as the host immune system, we can overcome key hurdles in cancer vaccine development.

## Figures and Tables

**Figure 1 vaccines-11-01783-f001:**
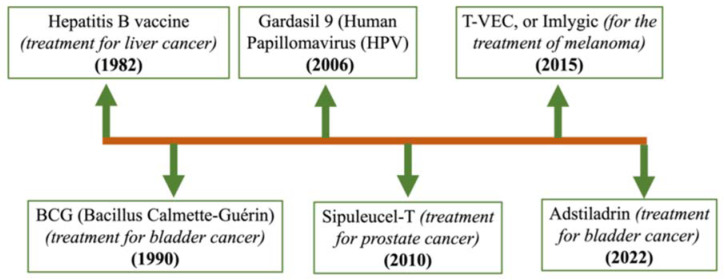
This academic picture presents a chronological timeline depicting the development/approval of several cancer vaccinations, with the developments arranged in a sequential manner.

**Figure 2 vaccines-11-01783-f002:**
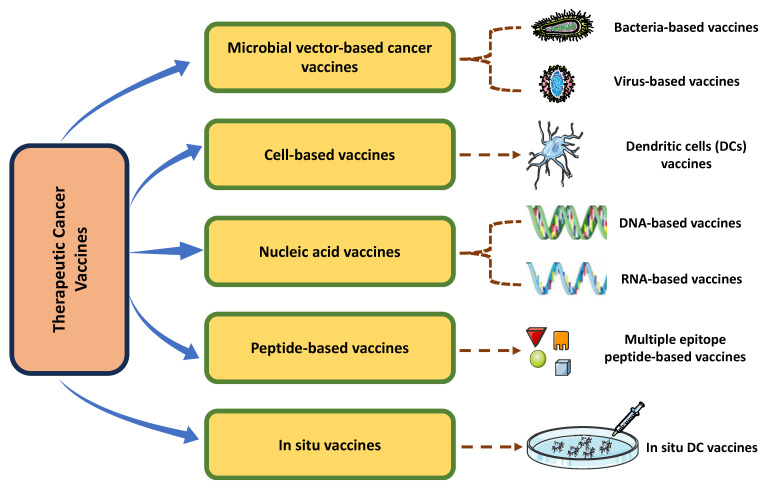
Overview of different types of targeted cancer vaccines. This figure was created using Servier Medical Art Commons Attribution 3.0 Unported Licence (https://smart.servier.com/).

**Table 1 vaccines-11-01783-t001:** Type of therapeutic vaccines/immunotherapies and mechanism.

Vaccine Method	Targets	Mechanism	Vaccine Type	Type of Cancer	Reference
Allogenic whole cancer cells with BCG stimulant	All tumor cell antigens	T cells, NK cells, macrophages, eosinophils	Cell-based	Prostate cancer, heterogenous metastatic melanoma	[36,37]
APC based vaccine	Tumor-associated antigen	Activation and maturation of DCs for CTL-based response	Cell-based	Metastatic prostate cancer	[38]
Adoptive T cell transfer	Tumor associated antigen	CTL-based response	Cell-based	Prostate cancer, metastatic melanomas	[39,40,41]
Proteins	Tumor-associated carbohydrates	Antibody development, T cell response	Antigen-based	Melanoma, lung cancer	[28,42]
Whole bacteria	Bacterial antigen	CTL and Ab response through immune stimulation	Antigen-based	Leukemia, bladder cancer, lung cancer, melanoma, and other neoplasms	[25]
DNA	Tumor associated antigen	CTL and Ab response	Antigen-based	Melanoma, mammary carcinoma, colon and lung carcinoma	[26,27,43]

**Table 2 vaccines-11-01783-t002:** Clinical trials of DNA vaccines in cancer care (https://clinicaltrials.gov).

ClinicalTrials.gov ID	Title	Phase	Condition	Treatment (s)
NCT02139267	A Randomized, Open-label, Multi-center, Phase 2 Clinical Trial to Determine the Optimal Dose and Evaluate the Safety of GX-188E, a DNA-based Therapeutic Vaccine, Administered Intramuscularly by Electroporation (EP) in HPV Type 16 and/or 18 Positive Patients with Cervical Intraepithelial Neoplasia 3 (CIN 3)	Phase 2	Cervical Intraepithelial Neoplasia	Biological: GX-188E
NCT04049864	Pilot Clinical Study of DNA Vaccination Against Neuroblastoma	Early Phase 1	Relapsed Neuroblastoma	Biological: DNA vaccine Biological: Salmonella oral vaccine Drug: Lenalidomide
NCT02529930	An Exploratory Safety and Immunogenicity Study of Human Papillomavirus (HPV16+) Immunotherapy VB10.16 in Women with High Grade Cervical Intraepithelial Neoplasia (HSIL; CIN 2/3)	Phase 1 Phase 2	High Grade Cervical Intraepithelial Neoplasia	Biological: VB10.16 Immunotherapy (DNA vaccine)
NCT03439085	A Phase 2, Open-Label Study to Evaluate Efficacy of Combination Treatment with MEDI0457 (INO-3112) and Durvalumab (MEDI4736) in Patients with Recurrent/Metastatic Human Papilloma Virus Associated Cancers	Phase 2	Human Papillomavirus-16 Positive Human Papillomavirus-18 Positive Metastatic Malignant Neoplasm Recurrent Anal Canal Carcinoma Recurrent Cervical Carcinoma Recurrent Malignant Neoplasm Recurrent Penile Carcinoma Recurrent Vaginal Carcinoma Recurrent Vulvar Carcinoma Refractory Malignant Neoplasm Stage IV Anal Cancer AJCC v8 Stage IV Cervical Cancer AJCC v8 Stage IV Penile Cancer AJCC v8 Stage IV Vaginal Cancer AJCC v8 Stage IV Vulvar Cancer AJCC v8 Stage IVA Cervical Cancer AJCC v8 Stage IVA Vaginal Cancer AJCC v8 Stage IVA Vulvar Cancer AJCC v8 Stage IVB Cervical Cancer AJCC v8 Stage IVB Vaginal Cancer AJCC v8 Stage IVB Vulvar Cancer AJCC v8	Biological: DNA Plasmid-encoding Interleukin-12/HPV DNA Plasmids Therapeutic Vaccine MEDI0457 Biological: Durvalumab
NCT00104845	Injection of AJCC Stage IIB, IIC, III, and IV Melanoma Patients with Human and Mouse gp100 DNA: A Phase I Trial to Assess Safety and Immune Response	Phase 1	Melanoma (Skin)	Biological: human gp100 plasmid DNA vaccine Biological: mouse gp100 plasmid DNA vaccine
NCT02348320	A Phase 1 Clinical Trial to Evaluate the Safety and Immunogenicity of a Personalized Polyepitope DNA Vaccine Strategy in Breast Cancer Patients with Persistent Triple-Negative Disease Following Neoadjuvant Chemotherapy	Phase 1	Triple Negative Breast Cancer	Biological: Personalized polyepitope DNA vaccine
NCT00199849	Safety and Immunological Evaluation of NY-ESO-1 Plasmid DNA (pPJV7611) Cancer Vaccine Given by Particle-mediated Epidermal Delivery (PMED) in Patients with Tumor Type Known to Express NY-ESO-1 or LAGE-1 Antigen.	Phase 1	Prostate Cancer Bladder Cancer Non-small Cell Lung Cancer Esophageal Cancer Sarcoma	Biological: NY-ESO-1 Plasmid DNA Cancer Vaccine

**Table 3 vaccines-11-01783-t003:** Clinical trials of RNA vaccines in cancer care (https://clinicaltrials.gov).

ClinicalTrials.gov ID	Title	Phase	Condition	Treatment(s)
NCT00626483	REGULATory T-Cell Inhibition with Basiliximab (Simulect^®^) During Recovery from Therapeutic Temozolomide-induced Lymphopenia During Antitumor Immunotherapy Targeted Against Cytomegalovirus in Patients with Newly Diagnosed Glioblastoma Multiforme	Phase 1	Malignant Neoplasms Brain	Biological: RNA-loaded dendritic cell vaccine Drug: Basiliximab
NCT05660408	Study of RNA-lipid Particle (RNA-LP) Vaccines for Recurrent Pulmonary Osteosarcoma (OSA)	Phase 1 Phase 2	Pulmonary Osteosarcoma	Biological: RNA-LP vaccine
NCT00108264	Tumor RNA Transfected Dendritic Cell Vaccines	Phase 1	Prostate Cancer	Biological: Tumor RNA transfected dendritic cells
NCT0348015	A Phase I/II Trial to Evaluate the Safety and Immunogenicity of a Messenger RNA (mRNA)-Based, Personalized Cancer Vaccine Against Neoantigens Expressed by the Autologous Cancer	Phase 1 Phase 2	Melanoma Colon Cancer Gastrointestinal Cancer Genitourinary Cancer Hepatocellular Cancer	Biological: National Cancer Institute (NCI)-4650, a messenger ribonucleic acid (mRNA)-based, Personalized Cancer Vaccine
NCT03418480	Therapeutic HPV Vaccine (BNT113) Trial in HPV16 Driven Carcinoma	Phase 1 Phase 2	Human Papilloma Virus Related Carcinoma Head and Neck Neoplasm Cervical Neoplasm Penile Neoplasms Malignant Unknown Primary Tumors	Drug: BNT113
NCT00004211	A Safety and Feasibility Study of Active Immunotherapy in Patients with Metastatic Prostate Carcinoma Using Autologous Dendritic Cells Pulsed with RNA Encoding Prostate Specific Antigen, PSA	Phase 1 Phase 2	Prostate Cancer	Biological: PSA RNA-pulsed dendritic cell vaccine

**Table 4 vaccines-11-01783-t004:** Clinical trials of peptide vaccines in cancer care (https://clinicaltrials.gov).

ClinicalTrials.gov ID	Title	Phase	Condition	Treatment(s)
NCT04509167	Pilot Study of Personalized Neoantigen Peptide Vaccines for the Treatment of Neoplasms	Early Phase 1	Neoplasms	Biological: Neoantigen Peptides
NCT05475106	Pilot Study of Personalized Neoantigen Peptide Vaccines and Leukine for the Treatment of Neoplasms	Early Phase 1	Neoplasms	Biological: Neoantigen Peptides
NCT00433745	Wilm’s Tumor 1 (WT1) Peptide Vaccination for Patients with High-Risk Hematological Malignancies	Phase 2	Myelodysplastic Syndrome Acute Myeloid Leukemia (AML) Chronic Myeloid Leukemia (CML)	Drug: WT1 Peptide Vaccine
NCT05013216	Mutant KRAS -Targeted Long Peptide Vaccine for Patients at High Risk of Developing Pancreatic Cancer	Phase 1	High Risk Cancer Pancreatic Cancer	Drug: KRAS peptide vaccine
NCT05741242	Basket Trial of Neoantigen Synthetic Long Peptide Vaccines in Patients with Advanced Malignancy	Phase 1 and Phase 2	Cancer Solid Tumor	Biological: Personalized Synthetic Long Peptide Vaccine
NCT00938223	This is an Open-label, Phase II Study of a Vaccine Comprising Melanoma Peptides, and a Tetanus Helper Peptide, Administered in GM-CSF-in-adjuvant. Patients Will be Randomized to Receive One of Two Different Vaccine Regimens. Patients Will be Stratified by Stage of Disease (IIB vs. III vs. IV).	Phase 2	Melanoma	Biological: 4-peptide and 12-peptide melanoma vaccines

**Table 5 vaccines-11-01783-t005:** Preclinical and clinical outcomes of combination therapy.

Combination Therapy	Outcomes	References
Dendritic cells and radiotherapy loaded with apoptotic heat-shock EC cell antigens.	This study had 40 participants. Upregulation in the expression of serum IFN-γ, IL-12, and IL-2, and the percentage of IFN-γ+ CD8+ T cells.	[134]
CRT coupled with multiple-epitope peptide vaccines.	Eleven patients with unresectable chemo-naïve ESCC showed peptide-specific cytotoxic lymphocyte responses to at least one of the five peptide antigens during vaccinations. Eight peptide vaccines plus CRT resulted in 54.5% of patients achieving CR and 45.5% experiencing programmed death (PD).	[135]
Systemic chemotherapy combined with DC for EC.	Five patients participated in this study. In primary tumors injected with labeled DC, this study showed that the DC accumulated but did not migrate to the lymph nodes.	[136]
Blockade of both the PD-1 and CTLA-4 checkpoints in addition to GVAX	In the CT26 murine model of colorectal cancer, resulted in 100% tumor rejection.	[129]
GVAX vaccine with PD-1/CTLA-4	IFN-γ+ TNF-α+ CD8+ tumor-infiltrating lymphocytes and CD8+/Treg ratios both significantly increased.	[137]
Peptide inhibitors of Foxp3 with murine tumor vaccine.	Foxp3 peptide inhibitors improved the anti-tumor effectiveness of a mouse tumor vaccination in two preclinical experiments.	[138]
Curcuminoids in combination with chemotherapy	In patients with solid tumors, such as gastric, colorectal, and breast cancer, this combination has shown greater efficacy.	[139]

## Data Availability

Not applicable.

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
