# Peer review of "Cancer Vaccines in the Immunotherapy Era: Promise and Potential"

_vaccines, 2023, doi:10.3390/vaccines11121783_

Round 1

Reviewer 1 Report

Comments and Suggestions for Authors

This short review titled “Cancer Vaccines in the Immunotherapy Era: Promise and Potential” provides a thorough examination of recent developments in research in therapeutic cancer vaccines. The review addresses several crucial aspects of cancer vaccine research, focusing on the promises and challenges of different vaccines in current clinical and preclinical studies, making it a timely contribution to the field.

This paper meets the publication requirements for Vaccines and is recommended for publication without reversion. 

Author Response

This short review titled “Cancer Vaccines in the Immunotherapy Era: Promise and Potential” provides a thorough examination of recent developments in research in therapeutic cancer vaccines. The review addresses several crucial aspects of cancer vaccine research, focusing on the promises and challenges of different vaccines in current clinical and preclinical studies, making it a timely contribution to the field.

This paper meets the publication requirements for Vaccines and is recommended for publication without reversion.

Response: We express our gratitude to the reviewer for thoroughly evaluating our work and recommending its publication without any necessary revisions.

Reviewer 2 Report

Comments and Suggestions for Authors

The review paper by Verma et al. represents a small slice through the immunotherapeutic approaches used in the cancer field. Very general report. In my humble opinion, what is missing is the selection path that suggests the selection of one or more types of immunotherapeutic vaccines to target tumor cells (cancer tumor stem cells vs. differentiated cancer cells?). Additionally, the text part about the peptide vaccine is too little. Overall, I did not get a message about how much time is needed to produce each type of vaccine or the effectiveness of each type of approach described in the text to target cold or hot tumors. Will neoantigen encoded by celluar miRNAs produce more value for immunotherapy than neoantigen encoded by processed celluar mRNA? At this point, I have very limited enthusiasm for the current review. Thank you

Author Response

The review paper by Verma et al. represents a small slice through the immunotherapeutic

approaches used in the cancer field. Very general report.

  1. In my humble opinion, what is missing is the selection path that suggests the selection of one or more types of immunotherapeutic vaccines to target tumor cells (cancer tumor stem cells vs. differentiated cancer cells?).

Response: Authors would like to take the opportunity to thank reviewer for this suggestion. We have made changes in the text from line number 136-160 (marked in red).

  1. Additionally, the text part about the peptide vaccine is too little.

Response: We thank this reviewer for his/her feedback and we find his/her suggestions very constructive for the improvement of this present manuscript. Authors have added text 240-271(marked in red).

  1. Overall, I did not get a message about how much time is needed to produce each type of vaccine or the effectiveness of each type of approach described in the text to target cold or hot tumors.

Response: We are thankful for your suggestions, and we have addressed all of your concerns and incorporated into revised manuscript. We have added text in the revised manuscript from line 347-355 (marked in red).

  1. Will neoantigen encoded by celluar miRNAs produce more value for immunotherapy than neoantigen encoded by processed celluar mRNA? At this point, I have very limited enthusiasm for the current review.

Response: We thank this reviewer for his/her feedback and we find his/her suggestions very constructive for the improvement of this present manuscript. Authors have added suggested changes in to text of revised manuscript line 373-386 (marked in red).

Reviewer 3 Report

Comments and Suggestions for Authors

The aim of this review is to discuss different types of cancer vaccines and their potential in treating cancers, mainly through immunotherapeutic approaches. While this review provide some useful insights in cancer vaccines, the information provided is rather scattered without logical connections. Specifically, the authors did not provide a systematic and comprehensive context of all the types of cancer vaccines as appeared in figure 1 and the link between these vaccines and their immunotherapeutic potential is weak. The authors should also address the following issues before this manuscript can be published.

1.     Authors need to have in-depth discussion of gene cancer vaccine (DNA, mRNA) and their non-viral delivery vehicles in section 1 and 2 to provide context before discussing their clinical landscape in section 3.

2.     This paper provides limited insights in the working mechanisms of cancer vaccines. The authors can improve this paper by adding more details in explaining how each type of cancer vaccine activate immune cells and lead to downstream interaction with tumors for eradication.

3.     Section two organization unclear. Section 2 should introduce the background of different types of cancer vaccines and discuss their working mechanisms, synthesis technique and pros and cons before proceeding to discuss their clinical applications in section 3. Shouldn’t section 2.3 to 2.5 assigned as sub-categories under section 2.2?

4.     The current contents in section 2.6 is mixed up and unclear as it mentions various types of vaccines (Bacteria, DNA, carbohydrates…) without providing enough context and discuss them in depth. It would be better to split up section 2.6 into subsections that has separate in depth discussion of antigens, protein/peptide, genes types of vaccines.

5.     Authors may consider adding timeline illustration of the development of different vaccines as some narratives are in chronicle order.

6.     Figure 1 should appear earlier to provide context to readers.

Comments on the Quality of English Language

Line 274, undefined abbreviation ICI. Please make sure that all abbreviations are defined before use.

Author Response

The aim of this review is to discuss different types of cancer vaccines and their potential in treating cancers, mainly through immunotherapeutic approaches. While this review provide some useful insights in cancer vaccines, the information provided is rather scattered without logical connections. Specifically, the authors did not provide a systematic and comprehensive context of all the types of cancer vaccines as appeared in figure 1 and the link between these vaccines and their immunotherapeutic potential is weak. The authors should also address the following issues before this manuscript can be published.

1, Authors need to have in-depth discussion of gene cancer vaccine (DNA, mRNA) and their non-viral delivery vehicles in section 1 and 2 to provide context before discussing their clinical landscape in section 3.

Response: We thank reviewer for this important feedback and we have added paragraph form 92-127 (marked in red).

2, This paper provides limited insights in the working mechanisms of cancer vaccines. The authors can improve this paper by adding more details in explaining how each type of cancer vaccine activate immune cells and lead to downstream interaction with tumors for eradication.

Response: We thank reviewer for this important feedback and we have changed this line as per reviewer’s suggestion (marked in red).

3, Section two organization unclear. Section 2 should introduce the background of different types of cancer vaccines and discuss their working mechanisms, synthesis technique and pros and cons before proceeding to discuss their clinical applications in section 3. Shouldn’t section 2.3 to 2.5 assigned as sub-categories under section 2.2?

Response: We express our gratitude to the reviewer for providing valuable feedback, and we have made the necessary revisions to the aforementioned line in accordance with the reviewer's comments. Also we have sub-categoried section 2.3 to 2.5 under section 2.2.

4, The current contents in section 2.6 is mixed up and unclear as it mentions various types of vaccines (Bacteria, DNA, carbohydrates…) without providing enough context and discuss them in depth. It would be better to split up section 2.6 into subsections that has separate in depth discussion of antigens, protein/peptide, genes types of vaccines.

Response: Authors thank reviewer for this important comment and we have added text marked in red from line 240-271.

5, Authors may consider adding timeline illustration of the development of different vaccines as some narratives are in chronicle order.

Response: We are in agreement with this reviewer and we have added figure 1 with the timeline illustration of the development of different vaccines in the revised manuscript.

6, Figure 1 should appear earlier to provide context to readers.

Response: Authors thank reviewer for this recommendation and we have made changes as per sugession about figure 1 (Now is figure 2).

Comments on the Quality of English Language

Line 274, undefined abbreviation ICI. Please make sure that all abbreviations are defined before use.

Response: Authors would like to take the opportunity to thank reviewer for this suggestion. We have made changes in the text (marked in red).

Round 2

Reviewer 3 Report

Comments and Suggestions for Authors

The authors have adequately addressed my comments.